# DISENTANGLING FACTORS OF VARIATION USING FEW LABELS

**Francesco Locatello**[1,2]**, Michael Tschannen**[3]**, Stefan Bauer**[2]**, Gunnar Rätsch**[1]**, Bernhard Schölkopf**[2]**, Olivier Bachem**[3]

[1] Department of Computer Science, ETH Zurich
[2] Max Planck Institute for Intelligent Systems, Tübingen
[3] Google Research, Brain Team
`francesco.locatello@inf.ethz.ch, bachem@google.com`

## ABSTRACT

Learning disentangled representations is considered a cornerstone problem in representation learning. Recently, Locatello et al. (2019) demonstrated that unsupervised disentanglement learning without inductive biases is theoretically impossible and that existing inductive biases and unsupervised methods do not allow to consistently learn disentangled representations. However, in many practical settings, one might have access to a limited amount of supervision, for example through manual labeling of (some) factors of variation in a few training examples. In this paper, we investigate the impact of such supervision on state-of-the-art disentanglement methods and perform a large scale study, training over $52\,000$ models under well-defined and reproducible experimental conditions. We observe that a small number of labeled examples (0.01–0.5% of the data set), with potentially imprecise and incomplete labels, is sufficient to perform model selection on state-of-the-art unsupervised models. Further, we investigate the benefit of incorporating supervision into the training process. Overall, we empirically validate that with little and imprecise supervision it is possible to reliably learn disentangled representations.

## 1 INTRODUCTION

In machine learning, it is commonly assumed that high-dimensional observations $\mathbf{x}$ (such as images) are the manifestation of a low-dimensional latent variable $\mathbf{z}$ of ground-truth factors of variation (Bengio et al., 2013; Kulkarni et al., 2015; Chen et al., 2016; Tschannen et al., 2018). More specifically, one often assumes that there is a distribution $p(\mathbf{z})$ over these latent variables and that observations in this ground-truth model are generated by sampling $\mathbf{z}$ from $p(\mathbf{z})$ first. Then, the observations $\mathbf{x}$ are sampled from a conditional distribution $p(\mathbf{x}|\mathbf{z})$. The goal of disentanglement learning is to find a representation of the data $r(\mathbf{x})$ which captures all the ground-truth factors of variation in $\mathbf{z}$ independently. The hope is that such representations will be interpretable, maximally compact, allow for counterfactual reasoning and be useful for a large variety of downstream tasks (Bengio et al., 2013; Peters et al., 2017; LeCun et al., 2015; Bengio et al., 2007; Schmidhuber, 1992; Lake et al., 2017; Goodfellow et al., 2009; Lenc & Vedaldi, 2015; Tschannen et al., 2018; Higgins et al., 2018; Suter et al., 2019; Adel et al., 2018; van Steenkiste et al., 2019; Locatello et al., 2019a; Gondal et al., 2019). As all these applications rely on the assumption that disentangled representations can be reliably learned in practice, we hope to learn them with as little supervision as possible (Bengio et al., 2013; Schölkopf et al., 2012; Peters et al., 2017; Pearl, 2009; Spirtes et al., 2000).

Current state-of-the-art unsupervised disentanglement approaches enrich the *Variational Autoencoder (VAE)* (Kingma & Welling, 2014) objective with different unsupervised regularizers that aim to encourage disentangled representations (Higgins et al., 2017a; Burgess et al., 2018; Kim & Mnih, 2018; Chen et al., 2018; Kumar et al., 2018; Rubenstein et al., 2018; Mathieu et al., 2018; Rolinek et al., 2019). The disentanglement of the representation of such methods exhibit a large variance and while some models turn out to be well disentangled it appears hard to identify them without supervision (Locatello et al., 2019b). This is consistent with the theoretical result of Locatello et al. (2019b) that the unsupervised learning of disentangled representations is impossible without inductive biases.

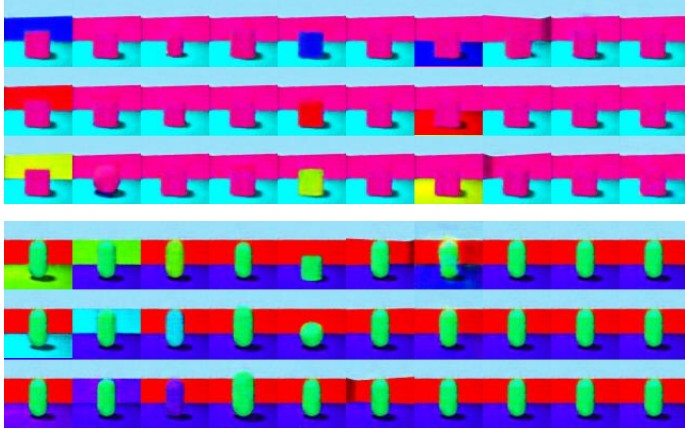

Figure 1: Latent traversals (each column corresponds to a different latent variable being varied) on Shapes3D for the $\beta$-TCVAE model with best validation MIG (top) and for the semi-supervised $\beta$-TCVAE model with best validation loss (bottom), both using only $1000$ labeled examples for validation and/or supervision. Both models appear to be visually well disentangled.

While human inspection can be used to select good model runs and hyperparameters (e.g. Higgins et al. (2017b, Appendix 5.1)), we argue that such supervision should be made explicit. We hence consider the setting where one has access to annotations (which we call *labels* in the following) of the latent variables $\mathbf{z}$ for a very limited number of observations $\mathbf{x}$, for example through human annotation. Even though this setting is not universally applicable (e.g. when the observations are not human interpretable) and a completely unsupervised approach would be elegant, collecting a small number of human annotations is simple and cheap via crowd-sourcing platforms such as Amazon Mechanical Turk, and is common practice in the development of real-world machine learning systems. As a consequence, the considered setup allows us to explicitly encode prior knowledge and biases into the learned representation via annotation, rather than relying solely on implicit biases such as the choice of network architecture with possibly hard-to-control effects.

Other forms of inductive biases such as relying on temporal information (video data) (Denton & Birodkar, 2017; Yingzhen & Mandt, 2018), allowing for interaction with the environment (Thomas et al., 2017), or incorporating grouping information (Kulkarni et al., 2015; Bouchacourt et al., 2018) were discussed in the literature as candidates to circumvent the impossibility result by Locatello et al. (2019b). However, there currently seems to be no quantitative evidence that such approaches will lead to improved disentanglement on the metrics and data sets considered in this paper. Furthermore, these approaches come with additional challenges: Incorporating time can significantly increase computational costs (processing video data) and interaction often results in slow training (e.g. a robot arm interacting with the real world). By contrast, collecting a few labels is cheap and fast.

We first investigate whether disentanglement scores are sample efficient and robust to imprecise labels. Second, we explore whether it is more beneficial to incorporate the limited amount of labels available into training and thoroughly test the benefits and trade-offs of this approach compared to supervised validation. For this purpose, we perform a reproducible large scale experimental study[1], training over $52\,000$ models on four different data sets. We found that unsupervised training with supervised validation enables reliable learning of disentangled representations. On the other hand, using some of the labeled data for training is beneficial both in terms of disentanglement and downstream performance. Overall, we show that a very small amount of supervision is enough to reliably learn disentangled representations as illustrated in Figure 1. Our key contributions can be summarized as follows:

- We observe that some of the existing disentanglement metrics (which require observations of $\mathbf{z}$) can be used to tune the hyperparameters of unsupervised methods even when only very few labeled examples are available (Section 3). Therefore, training a variety of models and introducing supervision to select the good runs is a viable solution to overcome the impossibility result of Locatello et al. (2019b).

---

[1]Reproducing these experiment requires approximately 8.57 GPU years (NVIDIA P100).

- We find that adding a simple supervised loss, using as little as 100 labeled examples, outperforms unsupervised training with supervised model validation both in terms of disentanglement scores and downstream performance (Section 4.2).

- We discover that both unsupervised training with supervised validation and semi-supervised training are surprisingly robust to label noise (Sections 3.2 and 4.3) and tolerate coarse and partial annotations, the latter being particularly important if not all factors of variation are known.

- Based on our findings we provide guidelines helpful for practitioners to leverage disentangled representations in practical scenarios.

## 2 BACKGROUND AND RELATED WORK

**Problem definition:** Consider a generative model with latent variable $\mathbf{z}$ with factorized density $p(\mathbf{z}) = \prod_{i=1}^{d} p(\mathbf{z}_i)$, where $d > 1$, and observations $\mathbf{x}$ obtained as samples from $p(\mathbf{x}|\mathbf{z})$. Intuitively, the goal of disentanglement learning is to find a representation $r(\mathbf{x})$ separating the factors of variation into independent components so that a change in a dimension of $\mathbf{z}$ corresponds to a change in a dimension of $r(\mathbf{x})$ (Bengio et al., 2013). Refinements of this definition include disentangling independent groups in the topological sense (Higgins et al., 2018) and learning disentangled causal models (Suter et al., 2019). These definitions are reflected in various disentanglement metrics that aim at measuring some structural property of the statistical dependencies between $\mathbf{z}$ and $r(\mathbf{x})$.

**Evaluating disentangled representations:** The *BetaVAE* (Higgins et al., 2017a) and *Factor-VAE* (Kim & Mnih, 2018) scores measures disentanglement by performing an intervention on the factors of variation and predicting which factor was intervened on. The *Mutual Information Gap (MIG)* (Chen et al., 2018), *Modularity* (Ridgeway & Mozer, 2018), *DCI Disentanglement* (Eastwood & Williams, 2018) and *SAPscore* (Kumar et al., 2018) first compute a matrix relating factors of variation and codes (for example via pairwise mutual information, feature importance and predictability). Then, they average a normalized difference between the top two entries of the matrix either row or column wise. For more details, see Appendix C of (Locatello et al., 2019b).

**Learning disentangled representations:** Since all these metrics require access to labels $\mathbf{z}$ they cannot be used for unsupervised training. Many state-of-the-art unsupervised disentanglement methods therefore extend VAEs (Kingma & Welling, 2014) with a regularizer $R_u(q_\phi(\mathbf{z}|\mathbf{x}))$ that enforces structure in the latent space of the VAE induced by the encoding distribution $q_\phi(\mathbf{z}|\mathbf{x})$ with the hope that this leads to disentangled representations. These approaches (Higgins et al., 2017a; Burgess et al., 2018; Kim & Mnih, 2018; Chen et al., 2018; Kumar et al., 2018) can be cast under the following optimization template:

$$\max_{\phi,\theta} \quad \underbrace{\mathbb{E}_{\mathbf{x}}[\mathbb{E}_{q_\phi(\mathbf{z}|\mathbf{x})}[\log p_\theta(\mathbf{x}|\mathbf{z})] - D_{\mathrm{KL}}(q_\phi(\mathbf{z}|\mathbf{x})\|p(\mathbf{z}))]}_{=:\mathrm{ELBO}(\phi,\theta)} + \beta\mathbb{E}_{\mathbf{x}}[R_u(q_\phi(\mathbf{z}|\mathbf{x}))]. \quad (1)$$

The $\beta$-VAE (Higgins et al., 2017a) and AnnealedVAE (Burgess et al., 2018) reduce the capacity of the VAE bottleneck under the assumption that encoding the factors of variation is the most efficient way to achieve a good reconstruction (Peters et al., 2017). The Factor-VAE (Kim & Mnih, 2018) and $\beta$-TCVAE both penalize the total correlation of the aggregated posterior $q(\mathbf{z})$ (i.e. the encoder distribution after marginalizing the training data). The DIP-VAE variants (Kumar et al., 2018) match the moments of the aggregated posterior and a factorized distribution. We refer to Appendix B of (Locatello et al., 2019b) and Section 3 of (Tschannen et al., 2018) for a more detailed description.

**Supervision and inductive biases:** While there is prior work on semi-supervised disentanglement learning (Reed et al., 2014; Cheung et al., 2014; Mathieu et al., 2016; Narayanaswamy et al., 2017; Kingma et al., 2014; Klys et al., 2018), these methods aim to disentangle only *some* observed factors of variation from the other latent variables which themselves remain entangled. These approaches are briefly discussed in Section 4. Exploiting relational information or knowledge of the effect of the factors of variation have both been qualitatively studied to learn disentangled representations (Hinton et al., 2011; Cohen & Welling, 2014; Karaletsos et al., 2015; Goroshin et al., 2015; Whitney et al., 2016; Fraccaro et al., 2017; Denton & Birodkar, 2017; Hsu et al., 2017; Yingzhen & Mandt, 2018; Locatello et al., 2018; Kulkarni et al., 2015; Ruiz et al., 2019; Bouchacourt et al., 2018; Adel et al., 2018).

**Other related work.** Due to the lack of a commonly accepted formal definition, the term "disentangled representations" has been used in very different lines of work. There is for example a rich

literature in disentangling pose from content in 3D objects and content from motion in videos (Yang et al., 2015; Yingzhen & Mandt, 2018; Hsieh et al., 2018; Fortuin et al., 2019; Deng et al., 2017; Goroshin et al., 2015; Hyvarinen & Morioka, 2016). This can be achieved with different degrees of supervision, ranging from fully unsupervised to semi-supervised. Another line of work aims at disentangling class labels from other latent variables by assuming the existence of a causal model where the latent variable $\mathbf{z}$ has an arbitrary factorization with the class variable $\mathbf{y}$. In this setting, $\mathbf{y}$ is partially observed (Reed et al., 2014; Cheung et al., 2014; Mathieu et al., 2016; Narayanaswamy et al., 2017; Kingma et al., 2014; Klys et al., 2018). Without further assumptions on the structure of the graphical model, this is equivalent to partially observed factors of variation with latent confounders. Except for very special cases, the recovery of the structure of the generative model is known to be impossible with purely observational data (Peters et al., 2017; D'Amour, 2019; Suter et al., 2019). Here, we intend to disentangle factors of variation in the sense of (Bengio et al., 2013; Suter et al., 2019; Higgins et al., 2018). We aim at separating the effects of all factors of variation, which translates to learning a representation with independent components. This problem has already been studied extensively in the non-linear ICA literature (Comon, 1994; Bach & Jordan, 2002; Jutten & Karhunen, 2003; Hyvarinen & Morioka, 2016; Hyvärinen & Pajunen, 1999; Hyvarinen et al., 2019; Gresele et al., 2019).

## 3  UNSUPERVISED TRAINING WITH SUPERVISED MODEL SELECTION

In this section, we investigate whether commonly used disentanglement metrics can be used to identify good models if a very small number of labeled observations is available. While existing metrics are often evaluated using as much as $10\,000$ labeled examples, it might be feasible in many practical settings to annotate 100 to 1000 data points and use them to obtain a disentangled representation. At the same time, it is unclear whether such an approach would work as existing disentanglement metrics have been found to be noisy (even with more samples) (Kim & Mnih, 2018). Finally, we emphasize that the impossibility result of Locatello et al. (2019b) does not apply in this setting as we do observe samples from $\mathbf{z}$.

### 3.1  EXPERIMENTAL SETUP AND APPROACH

**Data sets.** We consider four commonly used disentanglement data sets where one has explicit access to the ground-truth generative model and the factors of variation: *dSprites* (Higgins et al., 2017a), *Cars3D* (Reed et al., 2015), *SmallNORB* (LeCun et al., 2004) and *Shapes3D* (Kim & Mnih, 2018). Following (Locatello et al., 2019b), we consider the statistical setting where one directly samples from the generative model, effectively side-stepping the issue of empirical risk minimization and overfitting. For each data set, we assume to have either 100 or 1000 labeled examples available and a large amount of unlabeled observations. We note that 100 labels correspond to labeling 0.01% of dSprites, 0.5% of Cars3D, 0.4% of SmallNORB and 0.02% of Shapes3D.

**Perfect vs. imprecise labels.** In addition to using the perfect labels of the ground-truth generative model, we also consider the setting where the labels are imprecise. Specifically, we consider the cases were labels are *binned* to take at most five different values, are *noisy* (each observation of a factor of variation has 10% chance of being random) or *partial* (only two randomly drawn factors of variations are labeled). This is meant to simulate the trade-offs in the process of a practitioner quickly labeling a small number of images.

**Model selection metrics.** We use MIG (Chen et al., 2018), DCI Disentanglement (Eastwood & Williams, 2018) and SAP score (Kumar et al., 2018) for model selection as they can be used on purely observational data. In contrast, the BetaVAE (Higgins et al., 2017a) and FactorVAE (Kim & Mnih, 2018) scores cannot be used for model selection on observational data because they require access to the true generative model and the ability to perform interventions. At the same time, prior work has found all these disentanglement metrics to be substantially correlated (Locatello et al., 2019b).

**Experimental protocol.** In total, we consider 32 different experimental settings where an experimental setting corresponds to a data set (*dSprites*/*Cars3D*/*SmallNORB*/*Shapes3D*), a specific number of labeled examples (100/1000), and a labeling setting (perfect/binned/noisy/partial). For each considered setting, we generate five different sets of labeled examples using five different random seeds. For each of these labeled sets, we train cohorts of $\beta$-VAEs (Higgins et al., 2017a), $\beta$-TCVAEs (Chen et al., 2018), Factor-VAEs (Kim & Mnih, 2018), and DIP-VAE-Is (Kumar et al., 2018) where each

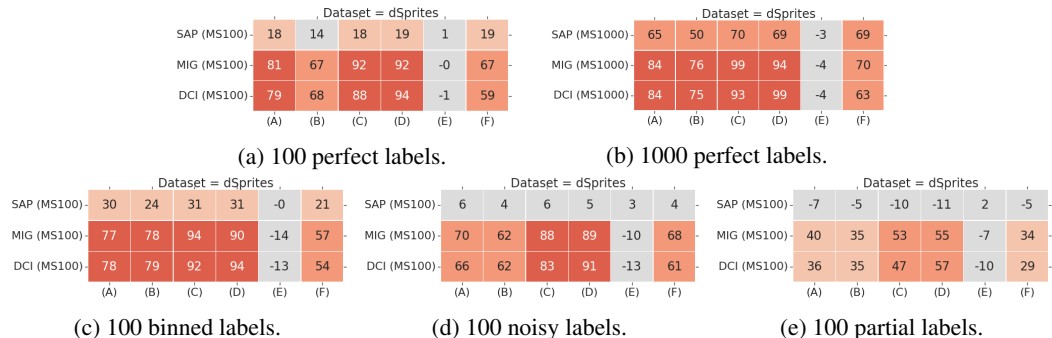

Figure 2: Rank correlation of validation metrics and test metrics on dSprites. Validation metrics are computed on different types of labels. Legend: (A)=BetaVAE Score, (B)=FactorVAE Score, (C)=MIG, (D)=DCI Disentanglement, (E)=Modularity, (F)=SAP.

model cohort consists of 36 different models with 6 different hyperparameters for each model and 6 random seeds. For a detailed description of hyperparameters, architecture, and model training we refer to Appendix B. For each of these 23 040 models, we then compute all the model selection metrics on the set of labeled examples and use these scores to select the best models in each of the cohorts. We prepend the prefix $U/S$ for *unsupervised* training with *supervised* model selection to the method name. Finally, we evaluate robust estimates of the BetaVAE score, the FactorVAE score, MIG, Modularity, DCI disentanglement and SAP score for each model based on an additional test set of 10 000 samples[2] in the same way as in Locatello et al. (2019b).

## 3.2 Key findings

We highlight our key findings with plots picked to be representative of our main results. In Appendices C–D, we provide complete sets of plots for different methods, data sets and metrics.

In Figure 2 (a), we show the rank correlation between the validation metrics computed on 100 samples and the test metrics on dSprites. We observe that MIG and DCI Disentanglement generally correlate well with the test metrics (with the only exception of Modularity) while the correlation for the SAP score is substantially lower. This is not surprising given that the SAP score requires us to train a multiclass support vector machine for each dimension of $r(\mathbf{x})$ predicting each dimension of $\mathbf{z}$. For example, on Cars3D the factor determining the object type can take 183 distinct values which can make it hard to train a classifier using only 100 training samples. In Figure 2 (b), we observe that the rank correlation improves considerably for the SAP score if we have 1000 labeled examples available and slightly for MIG and DCI Disentanglement. In Figure 1 (top) we show latent traversals for the $U/S$ model achieving maximum validation MIG on 1000 examples on Shapes3D. Figure 2 (c) shows the rank correlation between the model selection metrics with binned values and the test metrics with exact labels. We observe that the binned labeling does not seem detrimental to the performance of the model selection with few labels. We interpret these results as follows: For the purpose of disentanglement, fine-grained labeling is not critical as the different factors of variation can already be disentangled using coarse feedback. Interestingly, the rank correlation of the SAP score and the test metrics improves significantly (in particular for 100 labels). This is to be expected, as now we only have five classes for each factor of variation so the classification problem becomes easier and the estimate of the SAP score more reliable. In Figure 2 (d) we observe that noisy labels are only slightly impacting the performance. In Figure 2 (e), we can see that observing only two factors of variation still leads to a high correlation with the test scores, although the correlation is lower than for other forms of label corruption.

**Conclusions.** From this experiment, we conclude that it is possible to identify good runs and hyperparameter settings on the considered data sets using the MIG and the DCI Disentanglement based on 100 labeled examples. The SAP score may also be used, depending on how difficult the underlying classification problem is. Surprisingly, these metrics are reliable even if we do not collect the labels exactly. We conclude that labeling a small number of examples for supervised validation appears to be a reasonable solution to learn disentangled representations in practice. Not observing

---

[2]For the BetaVAE score and the FactorVAE score, this includes specific realizations based on interventions on the latent space.

all factors of variation does not have a dramatic impact. Whenever it is possible, it seems better to label more factors of variation in a coarser way rather than fewer factors more accurately.

## 4 INCORPORATING LABEL INFORMATION DURING TRAINING

Using labels for model selection—even only a small amount—raises the natural question whether these labels should rather be used for training a good model directly. In particular, such an approach also allows the structure of the ground-truth factors of variation to be used, for example ordinal information. In this section, we investigate a simple approach to incorporate the information of very few labels into existing unsupervised disentanglement methods and compare that approach to the alternative of unsupervised training with supervised model selection (as described in Section 3).

The key idea is that the limited labeling information should be used to ensure a latent space of the VAE with desirable structure w.r.t. the ground-truth factors of variation (as there is not enough labeled samples to learn a good representation solely from the labels). We hence incorporate supervision by equipping Equation 1 with a constraint $R_s(q_\phi(\mathbf{z}|\mathbf{x}), \mathbf{z}) \leq \kappa$, where $R_s(q_\phi(\mathbf{z}|\mathbf{x}), \mathbf{z})$ is a function computed on the (few) available observation-label pairs and $\kappa > 0$ is a threshold. We can now include this constraint into the loss as a regularizer under the Karush-Kuhn-Tucker conditions:

$$\max_{\phi, \theta} \quad \text{ELBO}(\phi, \theta) + \beta \mathbb{E}_\mathbf{x} R_u(q_\phi(\mathbf{z}|\mathbf{x})) + \gamma_{\text{sup}} \mathbb{E}_{\mathbf{x}, \mathbf{z}} R_s(q_\phi(\mathbf{z}|\mathbf{x}), \mathbf{z}) \qquad (2)$$

where $\gamma_{\text{sup}} > 0$. We rely on the binary cross-entropy loss to match the factors to their targets, i.e., $R_s(q_\phi(\mathbf{z}|\mathbf{x}), \mathbf{z}) = -\sum_{i=1}^d z_i \log(\sigma(r(\mathbf{x})_i)) + (1 - z_i) \log(1 - \sigma(r(\mathbf{x})_i))$, where the targets $z_i$ are normalized to $[0, 1]$, $\sigma(\cdot)$ is the logistic function and $r(\mathbf{x})$ corresponds to the mean (vector) of $q_\phi(\mathbf{z}|\mathbf{x})$. When $\mathbf{z}$ has more dimensions than the number of factors of variation, only the first $d$ dimensions are regularized (where $d$ is the number of factors of variation). While the $z_i$ do not model probabilities of a binary random variable but factors of variation with potentially more than two discrete states, we have found the binary cross-entropy loss to work empirically well out-of-the-box. We also experimented with a simple $L_2$ loss $\|\sigma(r(\mathbf{x})) - \mathbf{z}\|^2$ for $R_s$, but obtained significantly worse results than for the binary cross-entropy. Similar observations were made in the context of VAEs where the binary cross-entropy as reconstruction loss is widely used and outperforms the $L_2$ loss even when pixels have continuous values in $[0, 1]$ (see, e.g. the code accompanying Chen et al. (2018); Locatello et al. (2019b)). Many other candidates for supervised regularizers could be explored in future work. However, given the already extensive experiments in this study, this is beyond the scope of the present paper.

**Differences to prior work on semi-supervised disentanglement.** Existing semi-supervised approaches tackle the different problem of disentangling some factors of variation that are (partially) observed from the others that remain entangled (Reed et al., 2014; Cheung et al., 2014; Mathieu et al., 2016; Narayanaswamy et al., 2017; Kingma et al., 2014). In contrast, we assume to observe all ground-truth generative factors but only for a very limited number of observations. Disentangling only some of the factors of variation from the others is an interesting extension of this study. However, it is not clear how to adapt existing disentanglement scores to this different setup as they are designed to measure the disentanglement of *all* the factors of variation. We remark that the goal of the experiments in this section is to compare the two different approaches to incorporate supervision into state-of-the-art unsupervised disentanglement methods.

### 4.1 EXPERIMENTAL SETUP

**True vs. imprecise labels.** As in Section 3, we compare the effectiveness of the ground-truth labels with binned, noisy and partial labels on the performance of our semi-supervised approach. To understand the relevance of ordinal information induced by labels, we further explore the effect of randomly permuting the labels in Appendix A.

**Experimental protocol.** To include supervision during training we split the labeled examples in a 90%/10% train/validation split. We consider 40 different experimental settings each corresponding to a data set (*dSprites/Cars3D/SmallNORB/Shapes3D*), a specific number of labeled examples (100/1000), and a labeling setting (perfect/binned/noisy/partial/randomly permuted). We discuss the inductive biases of $R_s$ and report the analysis with randomly permuted labels in Appendix A. For each considered setting, we generate the same five different sets of labeled examples we used for the $U/S$ models. For each of the labeled sets, we train cohorts of $\beta$-VAEs, $\beta$-TCVAEs, Factor-VAEs, and

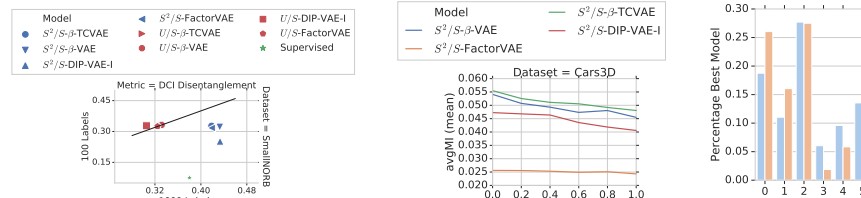

Figure 3: (left) Median across the draws of the labeled data set of the DCI Disentanglement test score on SmallNORB after validation with 100 and 1000 labeled examples. $U/S$ were validated with the MIG. (center) Increasing the supervised regularization strength makes the matrix of pairwise mutual information $I(\mathbf{z}, r(\mathbf{x}))$ closer to diagonal (avgMI $= \|I(\mathbf{z}, r(\mathbf{x})) - \text{diag}(I(\mathbf{z}, r(\mathbf{x})))\|_F^2$). (right) Probability of each method being the best on a random downstream task. Legend: 0=$S^2/S$-$\beta$-TCVAE, 1=$S^2/S$-$\beta$-VAE, 2=$S^2/S$-DIP-VAE-I, 3=$S^2/S$-FactorVAE, 4=$U/S$-$\beta$-TCVAE, 5=$U/S$-$\beta$-VAE, 6=$U/S$-DIP-VAE-I, 7=$U/S$-FactorVAE. For the $U/S$ methods we sample the validation metric uniformly.

DIP-VAE-Is with the additional supervised regularizer $R_s(q_\phi(\mathbf{z}|\mathbf{x}), \mathbf{z})$. Each model cohort consists of 36 different models with 6 different hyperparameters for each of the two regularizers and one random seed. Details on the hyperparameter values can be found in Appendix B. For each of these $28\,800$ models, we compute the value of $R_s$ on the validation examples and use these scores to select the best method in each of the cohorts. For these models we use the prefix $S^2/S$ for *semi-supervised* training with *supervised* model selection and compute the same test disentanglement metrics as in Section 3.

**Fully supervised baseline.** We further consider a fully supervised baseline where the encoder is trained solely based on the supervised loss (without any decoder, KL divergence and reconstruction loss) with perfectly labeled training examples (again with a $90\%/10\%$ train/validation split). The supervised loss does not have any tunable hyperparameter, and for each labeled data set, we run cohorts of six models with different random seeds. For each of these $240$ models, we compute the value of $R_s$ on the validation examples and use these scores to select the best method in the cohort.

### 4.2 SHOULD LABELS BE USED FOR TRAINING?

First, we investigate the benefit of including the label information during training by comparing semi-supervised training with supervised validation in Figure 3 (left). Each dot in the plot corresponds to the median of the DCI Disentanglement score across the draws of the labeled subset on SmallNORB (using 100 vs 1000 examples for validation). For the $U/S$ models we use MIG for validation (MIG has a higher rank correlation with most of the testing metrics than other validation metrics, see Figure 2). From this plot one can see that the fully supervised baseline performs worse than the ones that make use of unsupervised data. As expected, having more labels can improve the median downstream performance for the $S^2/S$ approaches (depending on the data set and the test metric) but does not improve the $U/S$ approaches (recall that we observed in Figure 2 (a) that the validation metrics already perform well with 100 samples).

To test whether incorporating the label information during training is better than using it for validation only, we report in Figure 4 (a) how often each approach outperforms all the others on a random disentanglement metric and data set. We observe that semi-supervised training often outperforms supervised validation. In particular, $S^2/S$-$\beta$-TC-VAE seems to improve the most, outperforming the $S^2/S$-Factor-VAE which was the best method for 100 labeled examples. Using 100 labeled examples, the $S^2/S$ approach already wins in 70.5% of the trials. In Appendix D, we observe similar trends even when we use the testing metrics for validation (based on the full testing set) in the $U/S$ models. The $S^2/S$ approach seem to overall improve training and to transfer well across the different disentanglement metrics. In Figure 1 (bottom) we show the latent traversals for the best $S^2/S$ $\beta$-TCVAE using 1000 labeled examples. We observe that it achieves excellent disentanglement and that the unnecessary dimensions of the latent space are unused, as desired.

In their Figure 27, Locatello et al. (2019b) showed that increasing regularization in unsupervised methods does not imply that the matrix holding the mutual information between all pairs of entries of $r(\mathbf{x})$ becomes closer to diagonal (which can be seen as a proxy for improved disentanglement). For the semi-supervised approach, in contrast, we observe in Figure 3 (center) that this is actually the case.

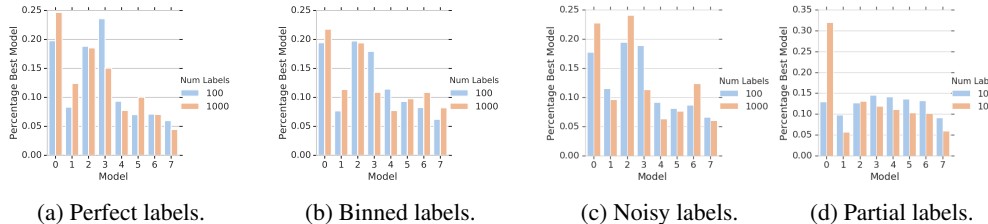

| (a) Perfect labels. | (b) Binned labels. | (c) Noisy labels. | (d) Partial labels. |

Figure 4: Probability of each method being the best on a random test metric and a random data set after validation with different types of labels. Legend: $0=S^2/S$-$\beta$-TCVAE, $1=S^2/S$-$\beta$-VAE, $2=S^2/S$-DIP-VAE-I, $3=S^2/S$-FactorVAE, $4=U/S$-$\beta$-TCVAE, $5=U/S$-$\beta$-VAE, $6=U/S$-DIP-VAE-I, $7=U/S$-FactorVAE. Overall, it seem more beneficial to incorporate supervision during training rather than using it only for validation. Having more labels available increases the gap.

Finally, we study the effect of semi-supervised training on the (natural) downstream task of predicting the ground-truth factors of variation from the latent representation. We use four different training set sizes for this downstream task: 10, 100, 1000 and 10 000 samples. We train the same cross-validated logistic regression and gradient boosting classifier as used in Locatello et al. (2019b). We observe in Figure 3 (right) that $S^2/S$ methods often outperform $U/S$ in downstream performance. From Figure 20 in the Appendix, one can see that the fully unsupervised method has often significantly worse performance.

**Conclusions:** Even though our semi-supervised training does not directly optimize the disentanglement scores, it seem beneficial compared to unsupervised training with supervised selection. The more labels are available the larger the benefit. Finding extremely sample efficient disentanglement metrics is however an important research direction for practical applications of disentanglement.

### 4.3 HOW ROBUST IS SEMI-SUPERVISED TRAINING TO IMPRECISE LABELS?

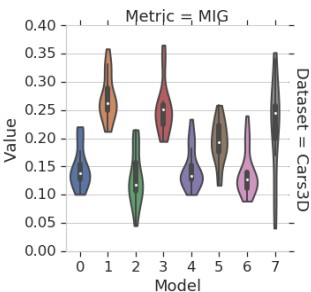

Figure 5: Distribution of models trained with different types of labels with 1000 samples, $U/S$ validated with MIG. Legend: $0=U/S$ perfect, $1=S^2/S$ perfect, $2=U/S$ binned, $3=S^2/S$ binned, $4=U/S$ noisy, $5=S^2/S$ noisy, $6=U/S$ partial, $7=S^2/S$ partial.

In this section, we explore the performance and the robustness of the $S^2/S$ methods compared to the $U/S$ methods. In Figure 5 we observe that imprecise labels do not significantly worsen the performance of both the supervised validation and the semi-supervised training. Sometimes the regularization induced by simplifying the labels appears to improve generalization, arguably due to a reduction in overfitting (see Figure 21 in the Appendix). We observe that the model selection metrics are slightly more robust than the semi-supervised loss. This effect is more evident especially when only 100 labeled examples are available, see Figure 21 in the Appendix. However, as shown in Figure 4 (b-d), the semi-supervised approaches still outperform supervised model selection in 64.8% and 67.5% of the trials with 100 binned and noisy labels respectively. The only exception appears to be with partial labels, where the two approaches are essentially equivalent (50.0%) with 100 labeled examples and the semi-supervised improves (62.6%) only with 1000 labeled examples.

**Conclusion:** These results show that the $S^2/S$ methods are also robust to imprecise labels. While the $U/S$ methods appear to be more robust, $S^2/S$ methods are still outperforming them.

## 5 CONCLUSION

In this paper, we investigated whether a very small number of labels can be sufficient to reliably learn disentangled representations. We found that existing disentanglement metrics can in fact be used to perform model selection on models trained in a completely unsupervised fashion even when the number of labels is very small and the labels are noisy. In addition, we showed that one can obtain even better results if one incorporates the labels into the learning process using a simple supervised regularizer. In particular, both unsupervised model selection and semi-supervised training are surprisingly robust to imprecise labels (inherent with human annotation) and partial

labeling of factors of variation (in case not all factors can be labeled), meaning that these approaches are readily applicable in real-world machine learning systems. The findings of this paper provide practical guidelines for practitioners to develop such systems and, as we hope, will help advancing disentanglement research towards more practical data sets and tasks.

**Acknowledgments:**   Francesco Locatello is supported by the Max Planck ETH Center for Learning Systems, by an ETH core grant (to Gunnar Rätsch), and by a Google Ph.D. Fellowship. This work was partially done while Francesco Locatello was at Google Research, Brain Team, Zurich.

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

| Method | Type | SAP 100 | SAP 1000 | MIG 100 | MIG 1000 | DCI 100 | DCI 1000 |
|---|---|---|---|---|---|---|---|
| $\beta$-VAE | $S^2/S$ permuted | 54.8% | 55.6% | 33.9% | 48.0% | 34.9% | 48.0% |
| | $U/S$ perfect | 45.2% | 44.4% | 66.1% | 52.0% | 65.1% | 52.0% |
| FactorVAE | $S^2/S$ permuted | 48.0% | 44.4% | 39.5% | 44.4% | 41.1% | 44.4% |
| | $U/S$ perfect | 52.0% | 55.6% | 60.5% | 55.6% | 58.9% | 55.6% |
| $\beta$-TCVAE | $S^2/S$ permuted | 64.8% | 55.5% | 34.4% | 42.3% | 36.8% | 43.5% |
| | $U/S$ perfect | 35.2% | 44.5% | 65.6% | 57.7% | 63.2% | 56.5% |
| DIP-VAE-I | $S^2/S$ permuted | 56.8% | 61.9% | 30.5% | 46.8% | 36.4% | 46.5% |
| | $U/S$ perfect | 43.2% | 38.1% | 69.5% | 53.2% | 63.6% | 53.5% |

Table 1: Removing the ordering information significantly worsen the performances of the $S^2/S$ on each method. The standard deviation is between 3% and 5% and can be computed as $\sqrt{p(1-p)/120}$.

## A  ORDERING AS AN INDUCTIVE BIAS

We emphasize that the considered supervised regularizer $R_s$ uses an inductive bias in the sense that it assumes the ordering of the factors of variation to matter. This inductive bias is valid for many ground truth factors of variation both in the considered data sets and the real world (such as spatial positions, sizes, angles or even color). We argue that such inductive biases should generally be exploited whenever they are available, which is the case if we have few manually annotated labels. To better understand the role of the ordinal information, we test its importance by removing it from the labels (via random permutation of the label order) before applying our semi-supervised approach. We find that this significantly degrades the disentanglement performance, i.e., ordinal information is indeed an important inductive bias. Note that permuting the label order should not harm the performance on the test metrics as they are invariant to permutations.

In this section, we verify that that the supervised regularizer we considered relies on the inductive bias given by the ordinal information present in the labels. Note that all the continuous factors of variation are binned in the considered data sets. We analyze how much the performance of the semi-supervised approach degrades when the ordering information is removed. To this end, we permute the order of the values of the factors of variation. Note that after removing the ordering information the supervised loss will still be at its minimum if $r(\mathbf{x})$ matches $\mathbf{z}$. However, the ordering information is now useless and potentially detrimental as it does not reflect the natural ordering of the true generative factors. We also remark that none of the disentanglement metrics make use of the ordinal information, so the performance degradation cannot be explained by fitting the wrong labels. In Figure 6, we observe that the $S^2/S$ approaches heavily rely on the ordering information and removing it significantly harms the performances of the test disentanglement metrics regardless of the fact that they are blind to ordering. This result is confirmed in Table 1, where we compute how often each $S^2/S$ method with permuted labels outperforms the corresponding $U/S$ on a random disentanglement metric and data set with perfect labels. We observe that in this case $U/S$ is superior most of the times but the gap reduces with more labels.

**Conclusions:** Imposing a suitable inductive bias (ordinal structure) on the ground-truth generative model in the form of a supervised regularizer is useful for disentanglement if the assumptions about the bias are correct. If the assumptions are incorrect, there is no benefit anymore over unsupervised training with supervised model selection (which is invariant to the ordinal structure).

## B  ARCHITECTURES AND DETAILED EXPERIMENTAL DESIGN

The architecture shared across every method is the default one in the `disentanglement_lib` which we describe here for completeness in Table 2 along with the other fixed hyperparameters in Table 4a and the discriminator for total correlation estimation in FactorVAE Table 4b with hyperparameters in Table 4c. The hyperparameters that were swept for the different methods can be found in Table 3. All the hyperparameters for which we report single values were not varied and are selected based on the literature.

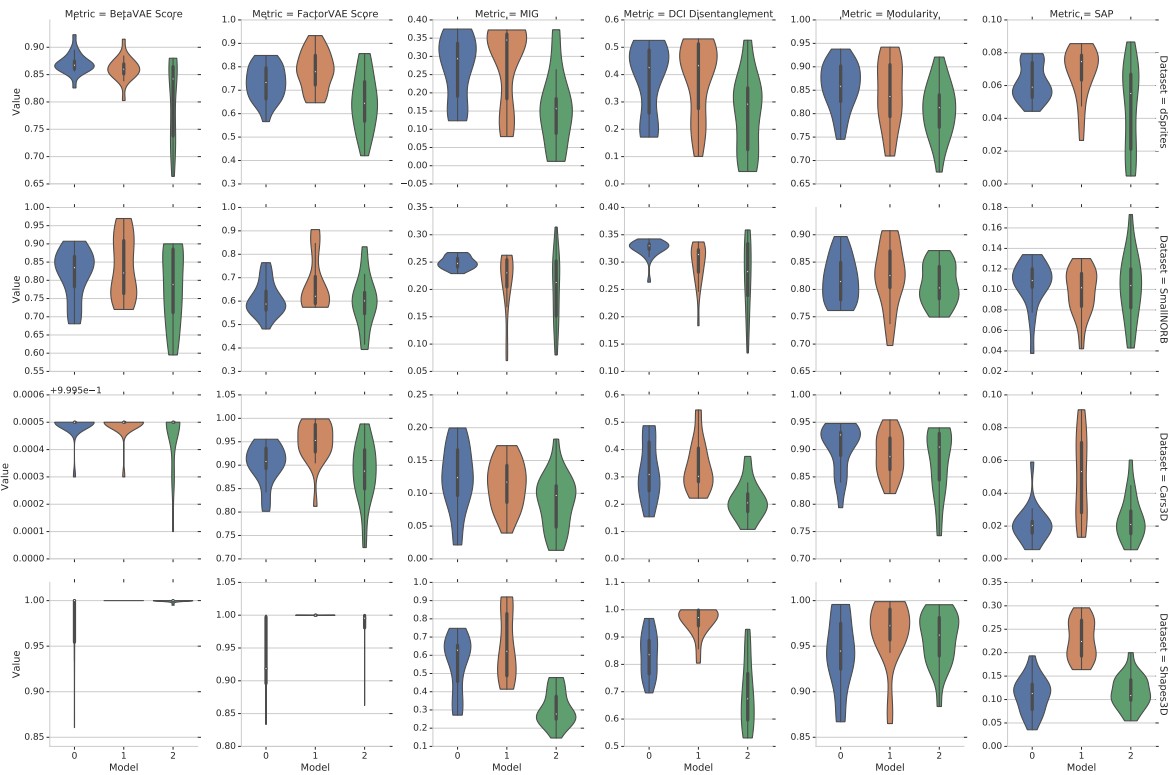

Figure 6: Violin plot showing the effect of removing the inductive bias given by the ordering of the labels on semi-supervised methods. Models are abbreviated as: $0=U/S$ with perfect labels, $1=S^2/S$ with perfect labels, $2=S^2/S$ training with permuted labels.

Table 2: Encoder and Decoder architecture for the main experiment.

| Encoder | Decoder |
|---|---|
| Input: $64 \times 64 \times$ number of channels | Input: $\mathbb{R}^{10}$ |
| $4 \times 4$ conv, 32 ReLU, stride 2 | FC, 256 ReLU |
| $4 \times 4$ conv, 32 ReLU, stride 2 | FC, $4 \times 4 \times 64$ ReLU |
| $2 \times 2$ conv, 64 ReLU, stride 2 | $4 \times 4$ upconv, 64 ReLU, stride 2 |
| $2 \times 2$ conv, 64 ReLU, stride 2 | $4 \times 4$ upconv, 32 ReLU, stride 2 |
| FC 256, FC $2 \times 10$ | $4 \times 4$ upconv, 32 ReLU, stride 2 |
| | $4 \times 4$ upconv, number of channels, stride 2 |

Table 3: Hyperparameters explored for the different disentanglement methods.

| Model | Parameter | Values |
|---|---|---|
| $\beta$-VAE | $\beta$ | [1, 2, 4, 6, 8, 16] |
| $S^2/S$ $\beta$-VAE | $\beta$ | [1, 2, 4, 6, 8, 16] |
| | $\gamma_{sup}$ | [1, 2, 4, 6, 8, 16] |
| FactorVAE | $\gamma$ | [10, 20, 30, 40, 50, 100] |
| $S^2/S$ FactorVAE | $\gamma$ | [10, 20, 30, 40, 50, 100] |
| | $\gamma_{sup}$ | [10, 20, 30, 40, 50, 100] |
| DIP-VAE-I | $\lambda_{od}$ | [1, 2, 5, 10, 20, 50] |
| | $\lambda_d$ | $10\lambda_{od}$ |
| $S^2/S$ DIP-VAE-I | $\lambda_{od}$ | [1, 2, 5, 10, 20, 50] |
| | $\lambda_d$ | $10\lambda_{od}$ |
| | $\gamma_{sup}$ | [1, 2, 5, 10, 20, 50] |
| $\beta$-TCVAE | $\beta$ | [1, 2, 4, 6, 8, 10] |
| $S^2/S$ $\beta$-TCVAE | $\beta$ | [1, 2, 4, 6, 8, 10] |
| | $\gamma_{sup}$ | [1, 2, 4, 6, 8, 10] |

Table 4: Other fixed hyperparameters.

(a) Hyperparameters common to all considered methods.

| Parameter | Values |
|---|---|
| Batch size | 64 |
| Latent space dimension | 10 |
| Optimizer | Adam |
| Adam: beta1 | 0.9 |
| Adam: beta2 | 0.999 |
| Adam: epsilon | 1e-8 |
| Adam: learning rate | 0.0001 |
| Decoder type | Bernoulli |
| Training steps | 300000 |

(b) Architecture for the discriminator in FactorVAE.

| Discriminator |
|---|
| FC, 1000 leaky ReLU |
| FC, 1000 leaky ReLU |
| FC, 1000 leaky ReLU |
| FC, 1000 leaky ReLU |
| FC, 1000 leaky ReLU |
| FC, 1000 leaky ReLU |
| FC, 2 |

(c) Parameters for the discriminator in FactorVAE.

| Parameter | Values |
|---|---|
| Batch size | 64 |
| Optimizer | Adam |
| Adam: beta1 | 0.5 |
| Adam: beta2 | 0.9 |
| Adam: epsilon | 1e-8 |
| Adam: learning rate | 0.0001 |

## C    DETAILED PLOTS FOR SECTION 3

In Figure 7, we compute the rank correlation between the validation metrics computed on 100 samples and the test metrics on each data set. In Figure 8, we observe that the correlation improves if we consider 1000 labeled examples. Figures 10 to 15 show the rank correlation between the validation metrics with binned/noisy/partial observations and the test metrics with exact labels for both 100 and 1000 examples. These plots are the extended version of Figure 2 showing the results on all data sets for both sample sizes.

In Figure 9, we plot for each unsupervised model its validation MIG with 100 samples against the DCI test score on dSprites. We can see that indeed there is a strong linear relationship.

## D    DETAILED PLOTS FOR SECTION 4

### D.1    DOES SUPERVISION HELP TRAINING?

In Figure 18 we plot the median of each score across the draws of the labeled subset achieved by the best models on each data set (using 100 vs 1000 examples). For the $U/S$ models we use MIG for validation (MIG has a higher rank correlation with most of the testing metric than other validation metrics, see Figure 2). This plot extends Figure 3 (left) to all data set and test score.

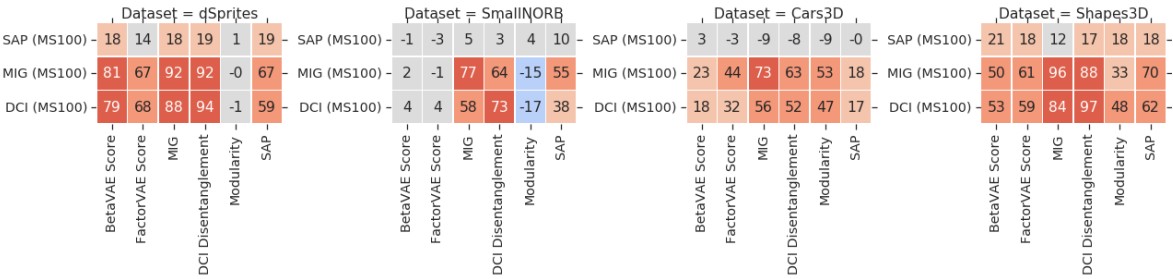

Figure 7: Rank correlation of validation metrics computed with 100 examples and test metrics on each data set.

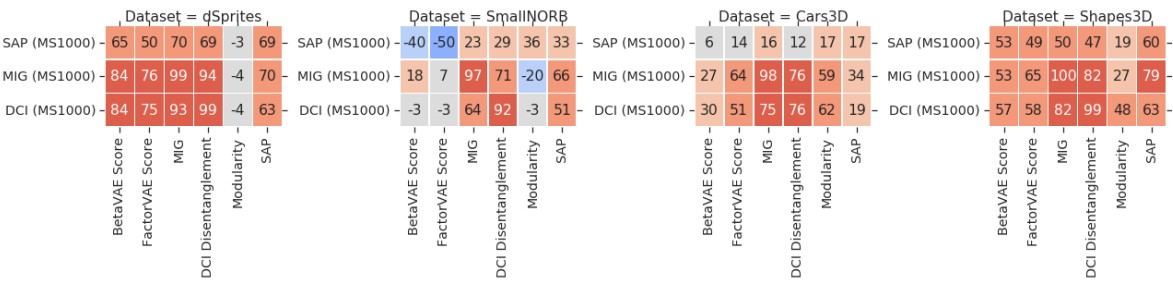

Figure 8: Rank correlation of validation metrics computed with 1000 examples and test metrics on each data set.

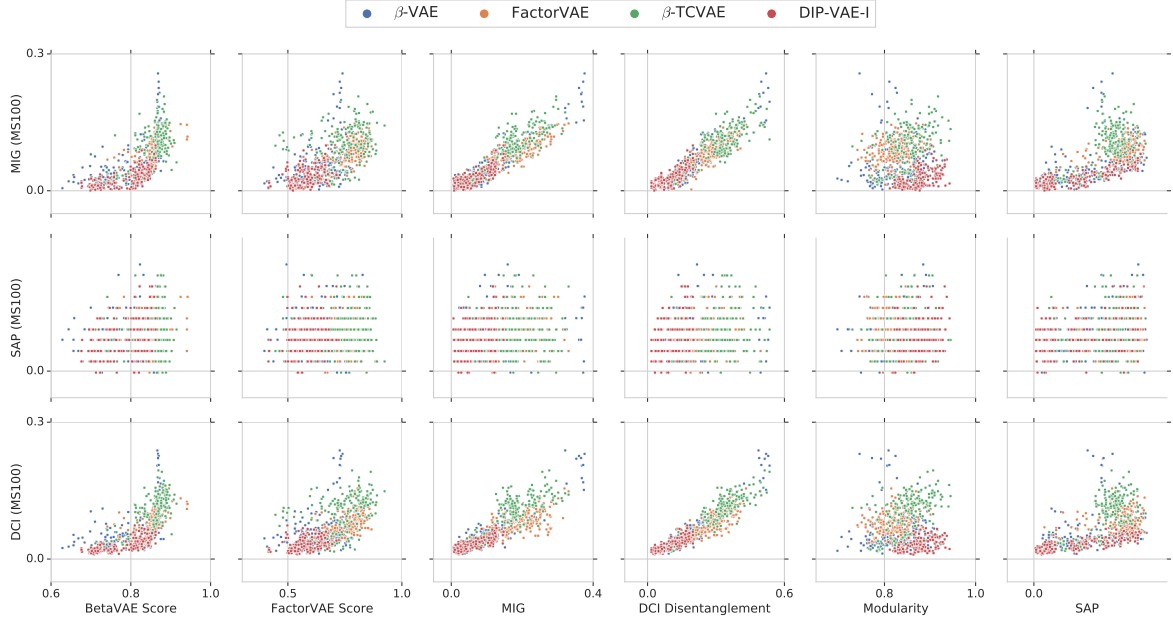

Figure 9: Scatter plot of validation and test metrics on dSprites before model selection. The validation metrics are computed with 100 examples.

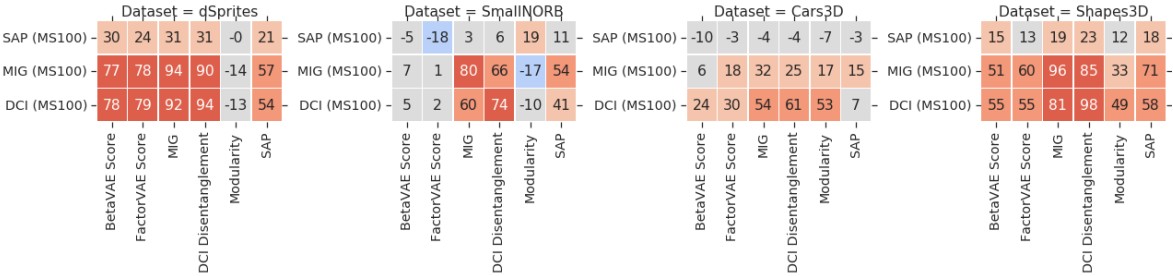

Figure 10: Rank correlation of validation metrics and test metrics. Validation metrics are computed with 100 examples with *labels binned to five categories*.

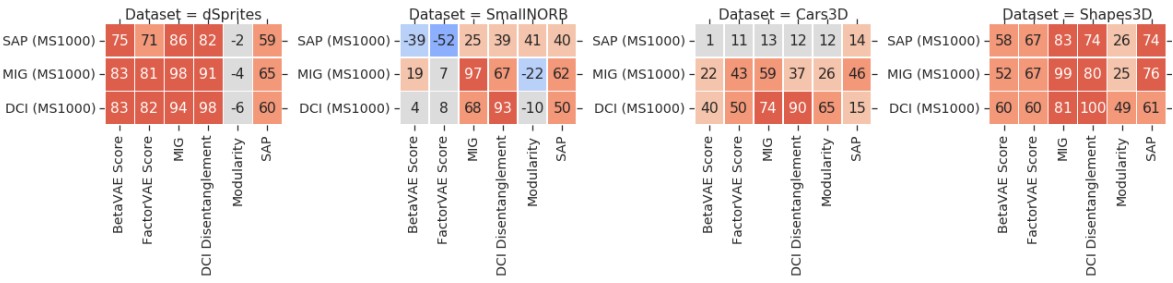

Figure 11: Rank correlation of validation metrics and test metrics. Validation metrics are computed with 1000 examples with *labels binned to five categories*.

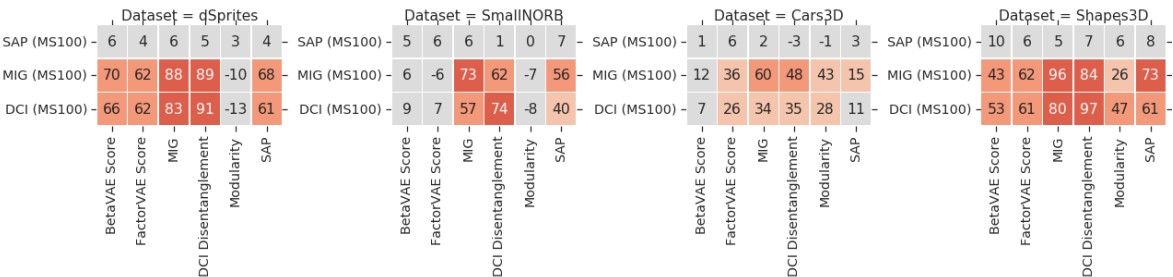

Figure 12: Rank correlation of validation metrics and test metrics. Validation metrics are computed with 100 examples where *each labeled factor has a 10% chance of being random*.

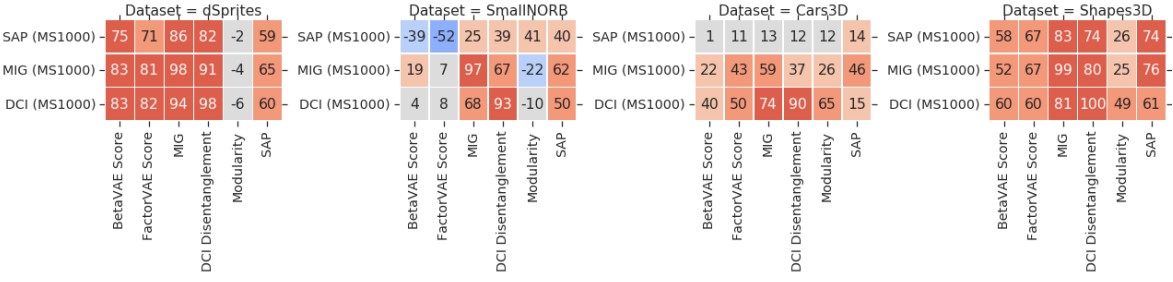

Figure 13: Rank correlation of validation metrics and test metrics. Validation metrics are computed with 1000 examples where *each labeled factor has a 10% chance of being random*.

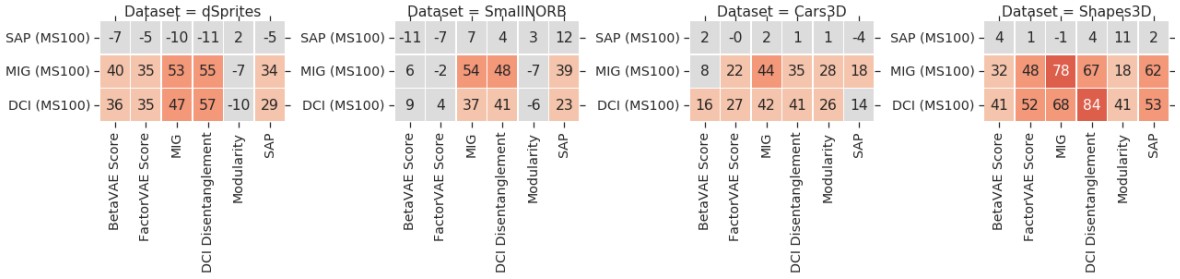

Figure 14: Rank correlation of validation metrics and test metrics. Validation metrics are computed with 100 examples with *only two factors labeled*.

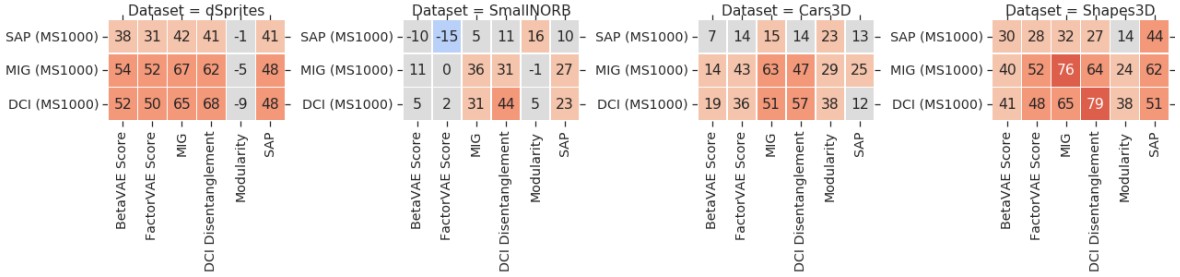

Figure 15: Rank correlation of validation metrics and test metrics. Validation metrics are computed with 1000 examples with *only two factors labeled*.

In Table 5, we compute how often each $S^2/S$ method outperforms the corresponding $U/S$ on a random disentanglement metric and data set. We observe that $S^2/S$ often outperforms $U/S$, especially when more labels are available.

In Figure 16 can be observed that with 1000 samples the semi-supervised method is often better than the corresponding $U/S$ even using the test MIG computed with 10 000 samples for validation. We conclude that the semi-supervised loss improves the training and transfer better to different metrics than the MIG. In Figure 17, we observe similar trends if we use the test DCI Disentanglement with 10 000 samples for validation of the $U/S$ methods.

in Figure 19 we observe that increasing the supervised regularization makes that the matrix holding the mutual information between all pairs of entries of $\mathbf{z}$ and $r(\mathbf{x})$ closer to diagonal. This plots extend Figure 3 to all data sets.

In Figure 20 we compare the median downstream performance after validation with 100 vs 1000 samples. Finally, we observe in Table 6 that semi-supervised methods often outperforms $U/S$ in downstream performance, especially when more labels are available.

### D.2 WHAT HAPPENS IF WE COLLECT IMPRECISE LABELS?

In Figures 21 and 22 we observe that imprecisions do not significantly worsen the performance of both the supervised validation and the semi-supervised training. These plots extend Figure 5 to both sample sizes, all test scores and data sets.

In Tables 7-9 we show how often each $S^2/S$ method outperforms the corresponding $U/S$ on a random disentanglement metric and data set with the different types of imprecise labels (binned/noisy/partial).

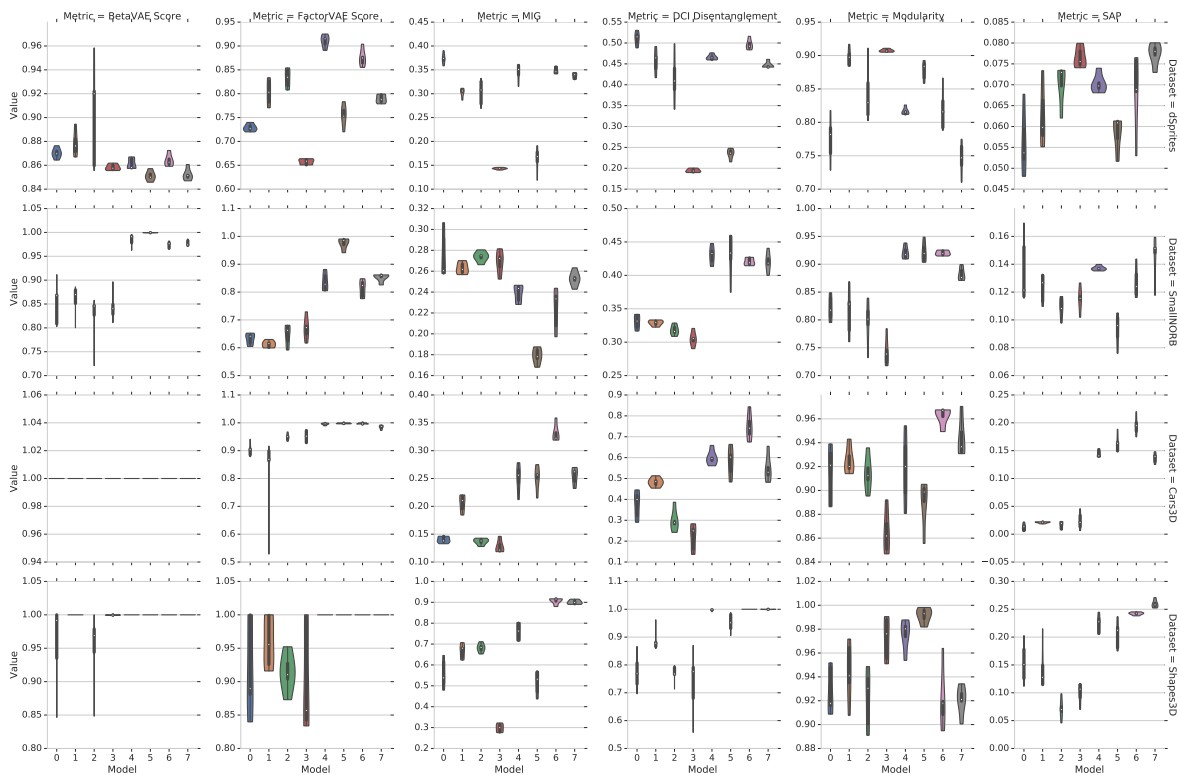

Figure 16: Test scores of the $U/S$ methods using the test MIG as validation and the $S^2/S$ models with 1000 labeled examples. Legend: 0=$U/S$-$\beta$-VAE, 1=$U/S$-$\beta$-TCVAE, 2=$U/S$-FactorVAE, 3=$U/S$-DIP-VAE-I, 4=$S^2/S$-$\beta$-VAE, 5=$S^2/S$-DIP-VAE-I, 6=$S^2/S$-$\beta$-TCVAE, 7=$S^2/S$-FactorVAE

| Method | Type | SAP 100 | SAP 1000 | MIG 100 | MIG 1000 | DCI 100 | DCI 1000 |
|--------|------|---------|----------|---------|----------|---------|----------|
| $\beta$-VAE | $S^2/S$ | 72.6% | 79.2% | 53.9% | 74.2% | 53.9% | 69.2% |
|  | $U/S$ | 27.4% | 20.8% | 46.1% | 25.8% | 46.1% | 30.8% |
| FactorVAE | $S^2/S$ | 71.5% | 79.4% | 64.5% | 75.2% | 68.5% | 77.6% |
|  | $U/S$ | 28.5% | 20.6% | 35.5% | 24.8% | 31.5% | 22.4% |
| $\beta$-TCVAE | $S^2/S$ | 79.5% | 80.6% | 58.5% | 75.0% | 62.9% | 74.4% |
|  | $U/S$ | 20.5% | 19.4% | 41.5% | 25.0% | 37.1% | 25.6% |
| DIP-VAE-I | $S^2/S$ | 81.6% | 83.5% | 64.9% | 74.8% | 67.7% | 70.5% |
|  | $U/S$ | 18.4% | 16.5% | 35.1% | 25.2% | 32.3% | 29.5% |

Table 5: Percentage of how often $S^2/S$ improves upon $U/S$ on for each approach separately. The standard deviation is between 3% and 5% and can be computed as $\sqrt{p(1-p)/120}$.

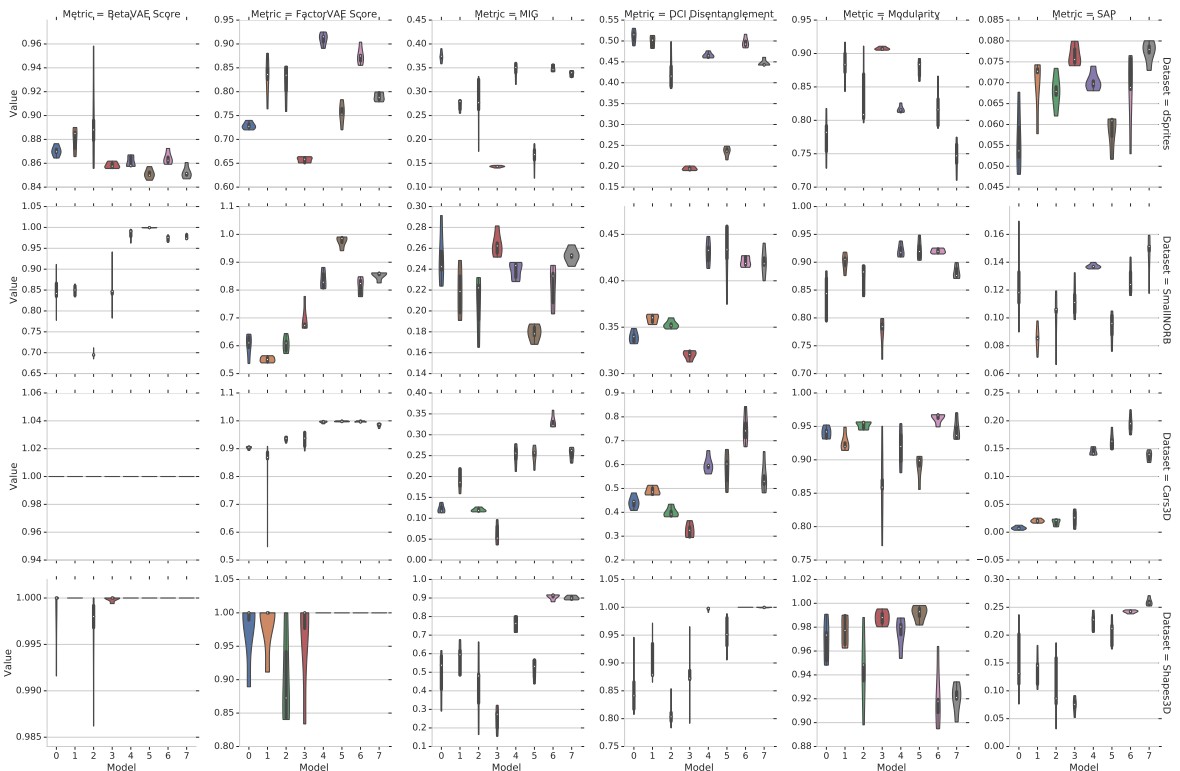

Figure 17: Test scores of the $U/S$ methods using the test DCI as validation and the $S^2/S$ models with 1000 labeled examples. Legend: 0=$U/S$-$\beta$-VAE, 1=$U/S$-$\beta$-TCVAE, 2=$U/S$-FactorVAE, 3=$U/S$-DIP-VAE-I, 4=$S^2/S$-$\beta$-VAE, 5=$S^2/S$-DIP-VAE-I, 6=$S^2/S$-$\beta$-TCVAE, 7=$S^2/S$-FactorVAE

| Method | Type | SAP 100 | SAP 1000 | MIG 100 | MIG 1000 | DCI 100 | DCI 1000 |
|---|---|---|---|---|---|---|---|
| $\beta$-VAE | $S^2/S$ | 70.0% | 75.6% | 53.8% | 75.0% | 43.8% | 71.9% |
| | $U/S$ | 30.0% | 24.4% | 46.2% | 25.0% | 56.2% | 28.1% |
| FactorVAE | $S^2/S$ | 61.2% | 71.9% | 62.5% | 78.1% | 63.8% | 71.2% |
| | $U/S$ | 38.8% | 28.1% | 37.5% | 21.9% | 36.2% | 28.8% |
| $\beta$-TCVAE | $S^2/S$ | 70.6% | 72.7% | 51.9% | 71.9% | 55.6% | 67.5% |
| | $U/S$ | 29.4% | 27.3% | 48.1% | 28.1% | 44.4% | 32.5% |
| DIP-VAE-I | $S^2/S$ | 71.9% | 80.6% | 50.6% | 75.6% | 50.6% | 65.0% |
| | $U/S$ | 28.1% | 19.4% | 49.4% | 24.4% | 49.4% | 35.0% |

Table 6: Percentage of how often $S^2/S$ improves upon $U/S$ on the downstream performance. The standard deviation is between 3% and 4% and can be computed as $\sqrt{p(1-p)/160}$.

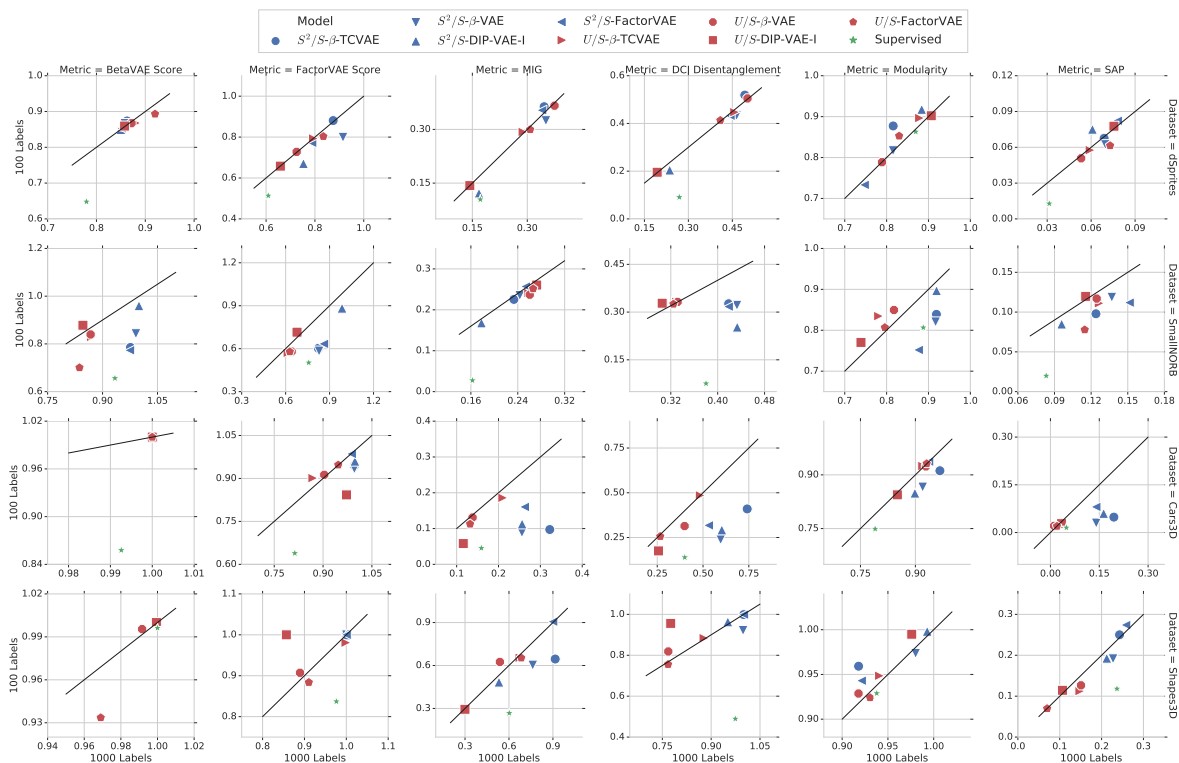

Figure 18: Median across the draws of the labeled data set of the test scores on each data set after validation with 100 and 1000 labeled examples. $U/S$ were validated with the MIG.

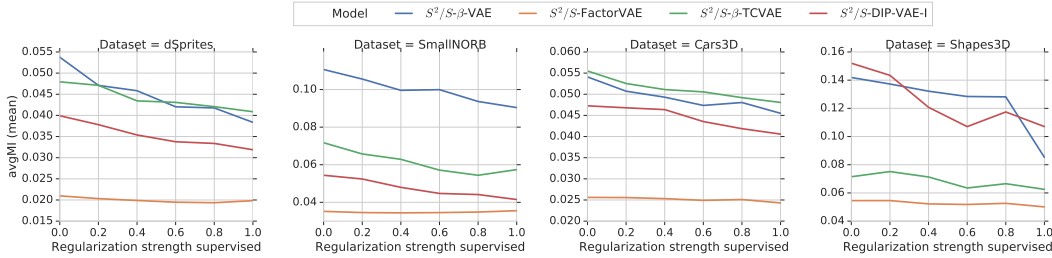

Figure 19: Increasing the supervised regularization strength makes the matrix of pairwise mutual information $I(\mathbf{z}, r(\mathbf{x}))$ closer to diagonal.

| Method | Type | SAP 100 | SAP 1000 | MIG 100 | MIG 1000 | DCI 100 | DCI 1000 |
|--------|------|---------|----------|---------|----------|---------|----------|
| $\beta$-VAE | $S^2/S$ | 66.9% | 76.3% | 50.4% | 75.2% | 44.5% | 74.6% |
|  | $U/S$ | 33.1% | 23.7% | 49.6% | 24.8% | 55.5% | 25.4% |
| FactorVAE | $S^2/S$ | 72.4% | 67.9% | 60.5% | 63.2% | 56.8% | 62.4% |
|  | $U/S$ | 27.6% | 32.1% | 39.5% | 36.8% | 43.2% | 37.6% |
| $\beta$-TCVAE | $S^2/S$ | 79.2% | 77.9% | 58.5% | 74.0% | 61.2% | 72.7% |
|  | $U/S$ | 20.8% | 22.1% | 41.5% | 26.0% | 38.8% | 27.3% |
| DIP-VAE-I | $S^2/S$ | 67.7% | 75.8% | 57.4% | 71.8% | 53.4% | 69.6% |
|  | $U/S$ | 32.3% | 24.2% | 42.6% | 28.2% | 46.6% | 30.4% |

Table 7: Percentage of how often $S^2/S$ improves upon $U/S$ for each method on a random disentanglement score and data set with binned labels. The standard deviation is between 3% and 5% and can be computed as $\sqrt{p(1-p)/120}$.

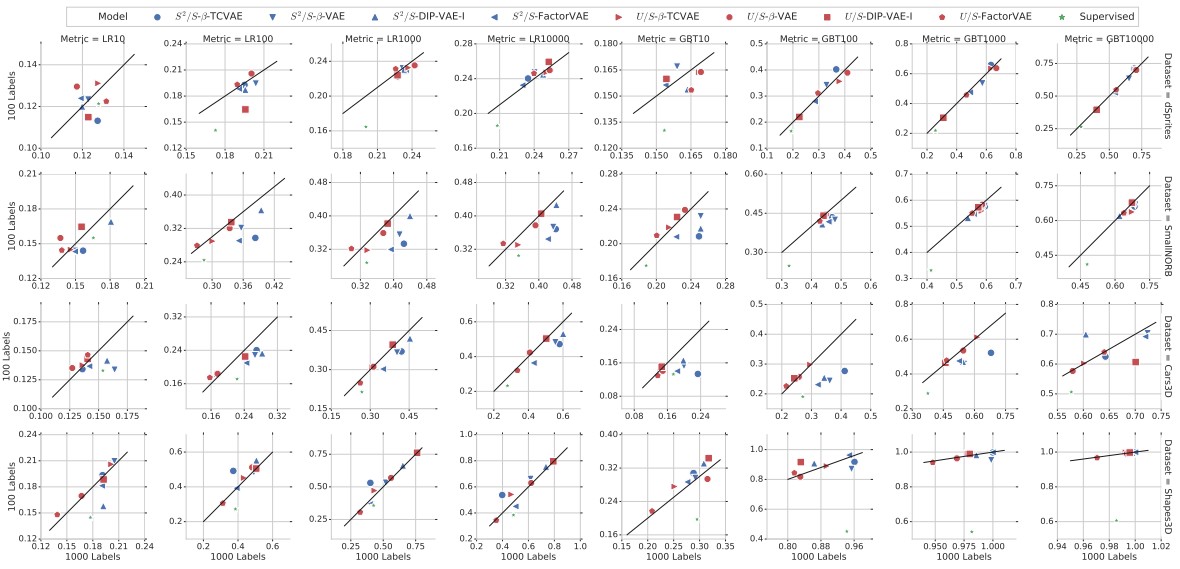

Figure 20: Comparison of the median downstream performance after validation with 100 vs. 1000 examples on each data set. The downstream tasks are: cross-validated Logistic Regression (LR) and Gradient Boosting classifier (GBT) both trained with 10, 100, 1000 and 10 000 examples.

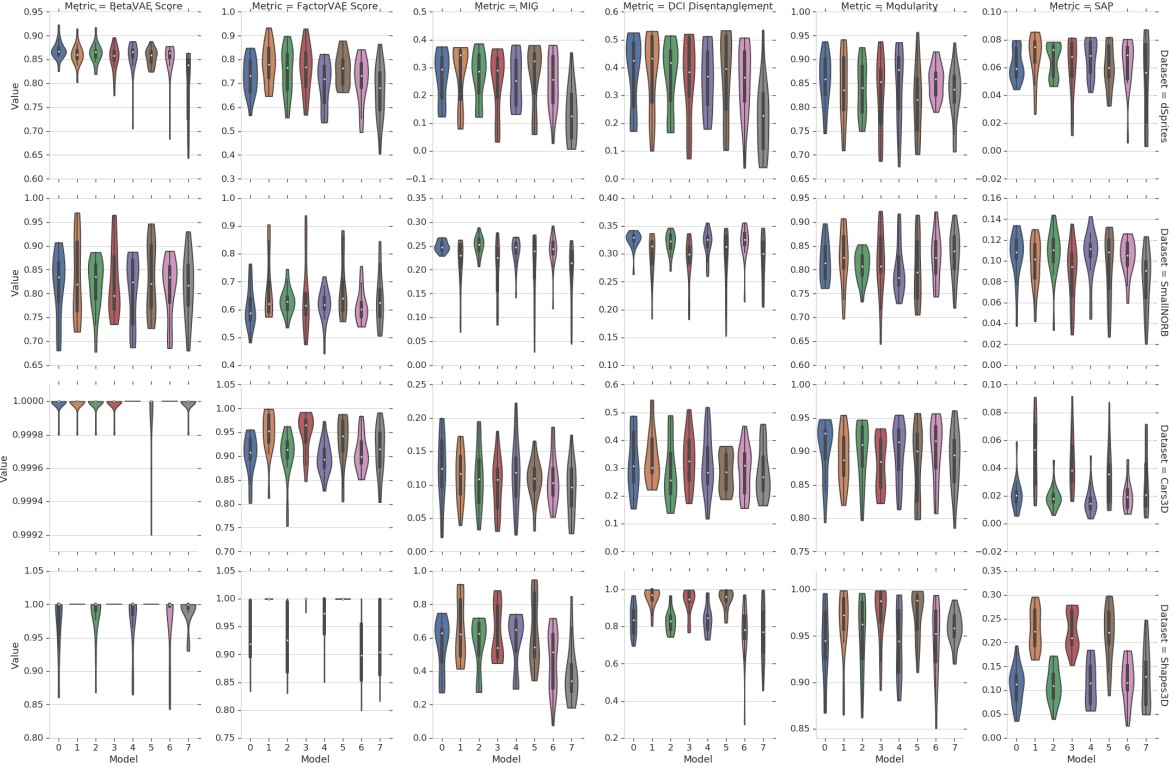

Figure 21: Distribution of models trained with perfect and imprecise labels with 100 samples, $U/S$ validated with MIG. Legend: 0=$U/S$ perfect, 1=$S^2/S$ perfect, 2=$U/S$ binned, 3=$S^2/S$ binned, 4=$U/S$ noisy, 5=$S^2/S$ noisy, 6=$U/S$ partial, 7=$S^2/S$ partial.

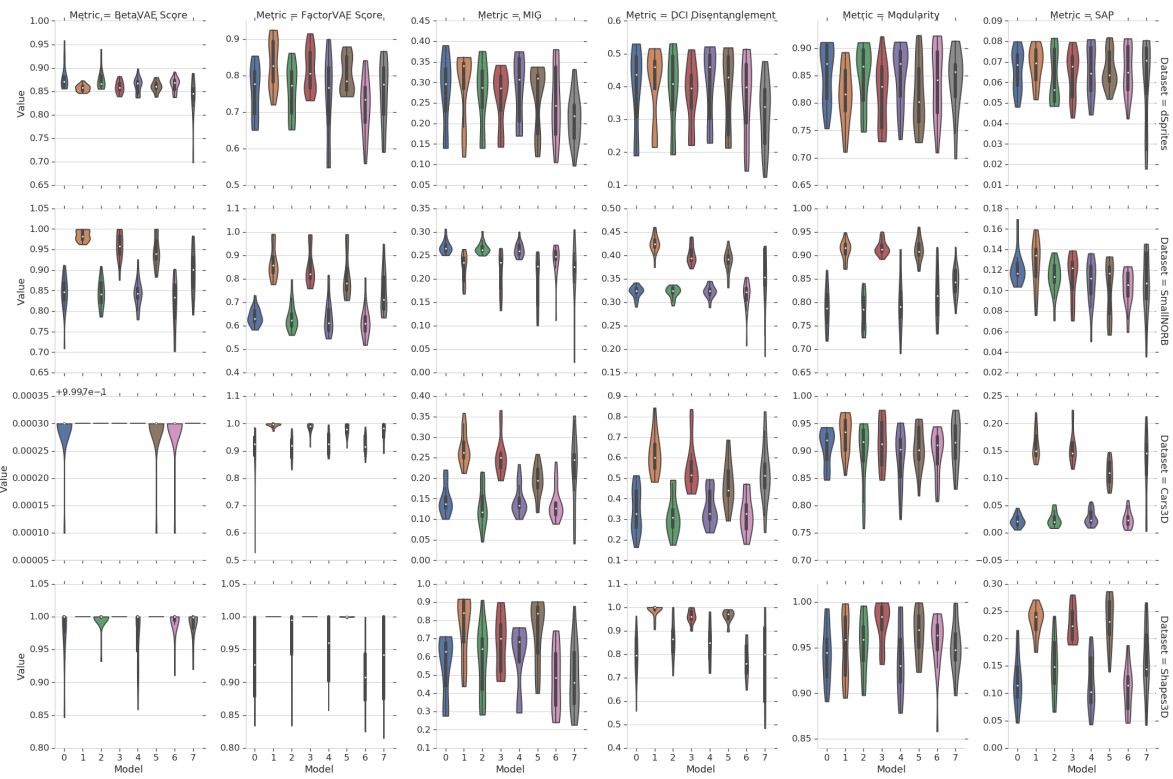

Figure 22: Distribution of models trained with perfect and imprecise labels with 1000 samples, $U/S$ validated with MIG. Legend: $0=U/S$ perfect, $1=S^2/S$ perfect, $2=U/S$ binned, $3=S^2/S$ binned, $4=U/S$ noisy, $5=S^2/S$ noisy, $6=U/S$ partial, $7=S^2/S$ partial.

| Method | Type | SAP 100 | SAP 1000 | MIG 100 | MIG 1000 | DCI 100 | DCI 1000 |
|---|---|---|---|---|---|---|---|
| $\beta$-VAE | $S^2/S$ | 71.0% | 80.2% | 49.6% | 70.5% | 53.5% | 72.1% |
| | $U/S$ | 29.0% | 19.8% | 50.4% | 29.5% | 46.5% | 27.9% |
| FactorVAE | $S^2/S$ | 70.9% | 75.4% | 63.2% | 67.5% | 64.8% | 64.6% |
| | $U/S$ | 29.1% | 24.6% | 36.8% | 32.5% | 35.2% | 35.4% |
| $\beta$-TCVAE | $S^2/S$ | 81.3% | 82.7% | 63.3% | 72.9% | 52.3% | 73.5% |
| | $U/S$ | 18.7% | 17.3% | 36.7% | 27.1% | 47.7% | 26.5% |
| DIP-VAE-I | $S^2/S$ | 79.5% | 77.2% | 53.0% | 63.7% | 51.9% | 62.2% |
| | $U/S$ | 20.5% | 22.8% | 47.0% | 36.3% | 48.1% | 37.8% |

Table 8: Percentage of how often $S^2/S$ improves upon $U/S$ for each method on a random disentanglement score and data set with noisy labels. The standard deviation is between 3% and 5% and can be computed as $\sqrt{p(1-p)/120}$.

| Method | Type | SAP 100 | SAP 1000 | MIG 100 | MIG 1000 | DCI 100 | DCI 1000 |
|---|---|---|---|---|---|---|---|
| $\beta$-VAE | $S^2/S$ | 62.7% | 65.6% | 34.1% | 47.2% | 45.6% | 56.0% |
| | $U/S$ | 37.3% | 34.4% | 65.9% | 52.8% | 54.4% | 44.0% |
| FactorVAE | $S^2/S$ | 59.2% | 68.8% | 46.4% | 63.7% | 50.4% | 63.2% |
| | $U/S$ | 40.8% | 31.2% | 53.6% | 36.3% | 49.6% | 36.8% |
| $\beta$-TCVAE | $S^2/S$ | 69.6% | 75.8% | 37.0% | 68.8% | 49.2% | 66.9% |
| | $U/S$ | 30.4% | 24.2% | 63.0% | 31.2% | 50.8% | 33.1% |
| DIP-VAE-I | $S^2/S$ | 59.2% | 72.8% | 48.4% | 61.6% | 41.4% | 55.5% |
| | $U/S$ | 40.8% | 27.2% | 51.6% | 38.4% | 58.6% | 44.5% |

Table 9: Percentage of how often $S^2/S$ improves upon $U/S$ for each method on a random disentanglement score and data set with partial labels (only two factors observed). The standard deviation is between 3% and 5% and can be computed as $\sqrt{p(1-p)/120}$.

