# OpenReview forum: "Disentangling Factors of Variations Using Few Labels"
_ICLR.cc/2020/Conference — Accept (Poster)_

### Official Review · AnonReviewer1 · 2019-10-15
**Official Blind Review #1**

**Rating:** 1

**Review:**

After rebuttal edit:
No clarifications were made, so I keep my score as is.

------------------------------------------------------
Claims: Explicitly requiring a small number of labels allows for successful learning of disentangled features by either using them as a validation set for hyper parameter tuning, or using them as a supervised loss.

Decision: Reject. This paper needs a substantial rewrite to make clear what specific contributions are from the multitude of experiments run in this study. As is, the two contributions stated in the introduction are both obvious and not particularly significant -- that having some labels of the type of disentanglement desired helps when used as a validation set and as a small number of labels for learning a disentangled representation space. There are no obviously stated conclusions about which types of labels are better than others (4.2). Section 3.2 seems to have some interesting findings that small scale supervision can help significantly and fine-grained labeling is not necessarily needed, but I don't understand why that finding is presented there when Fig. 4 seems to perform a similar experiment on types of labels with no conclusion based on its results. Conclusion sentence of 4.3 is hard to decipher, but I assume is just saying S^2/S beats U/S even when S^2/S is subject to noisy labels. Overall, I find it very difficult to absorb the huge amount of results and find the analysis not well presented.



**Experience Assessment:**

I do not know much about this area.

**Review Assessment: Checking Correctness Of Derivations And Theory:**

N/A

**Review Assessment: Checking Correctness Of Experiments:**

I assessed the sensibility of the experiments.

**Review Assessment: Thoroughness In Paper Reading:**

I read the paper at least twice and used my best judgement in assessing the paper.

---

> ### Author Response · Authors · 2019-11-08
> **Answer to Reviewer 1**
>
> We respectfully disagree with the reviewer's assessment. The study is clearly structured into two logical parts (semi-supervised model selection vs semi-supervised training) and contains tangible conclusions/insights at the end of each experiment section, in the introduction, and overall conclusion. This holds in particular for the sections 3.2, 4.2, and 4.3 mentioned by the reviewer. While the overall idea that supervision is useful for disentanglement may not be particularly surprising, sections 3.2, 4.2, and 4.3 contain detailed and useful insights, based on experimental evidence, about how supervision is best used.

---

> > ### Author Response · Authors · 2019-11-14
> > **Follow-up answer to Reviewer 1**
> >
> > GENERAL COMMENT:
> > We disagree with the reviewer's post-rebuttal response that "no clarifications were made". The reviewer's main concern appears to be "This paper needs a substantial rewrite to make clear what specific contributions are from the multitude of experiments run in this study." We strongly disagree with this (non-constructive) concern and have linked in our rebuttal to the relevant section in the paper where such conclusions are explicitly stated. To illustrate this point, consider the ad-verbatim passages from the manuscript which directly refute the reviewer's concern:
> >
> > INTRODUCTION (bullet points):
> > * “We observe that some of the existing disentanglement metrics (which require observations of z) can be used to tune the hyperparameters of unsupervised methods even when only very few labeled examples are available (Section 3). [...]”.
> >
> > * “We find that adding a simple supervised loss, using as little as 100 labeled examples, outperforms unsupervised training with supervised model validation both in terms of disentanglement scores and downstream performance (Section 4.2)”.
> >
> > * “Both unsupervised training with supervised validation and semi-supervised training are surprisingly robust to label noise (Sections 3.2 and 4.3) and tolerate coarse and partial annotations [...]”.
> >
> > * “Based on our findings we provide guidelines helpful for practitioners to leverage disentangled representations in practical scenarios.”
> >
> > CONCLUSIONS OF SECTION 3.2:
> > “From this experiment, we conclude that it is possible to identify good runs and hyperparameter settings on the considered data sets using the MIG and the DCI Disentanglement based on 100 labeled examples. [...] Not observing all factors of variation does not have a dramatic impact. Whenever it is possible, it seems better to label more factors of variation in a coarser way rather than fewer factors more accurately.”
> >
> > CONCLUSIONS OF SECTION 4.2 (SHOULD LABELS BE USED FOR TRAINING?):
> > “Even though our semi-supervised training does not directly optimize the disentanglement scores, it seem beneficial compared to unsupervised training with supervised selection. The more labels are available the larger the benefit. [...]“
> >
> > CONCLUSIONS OF SECTION 4.3  (HOW ROBUST IS SEMI-SUPERVISED TRAINING TO IMPRECISE LABELS?):
> > “These results show that the $S^2/S$ are also robust to imprecise labels. While the $U/S$ methods appear to be more robust, $S^2/S$ methods are still outperforming them.”
> >
> > OVERALL CONCLUSIONS OF THE PAPER:
> > “In this paper, we investigated whether a very small number of labels can be sufficient to reliably learn disentangled representations. We found that existing disentanglement metrics can in fact be used to perform model selection on models trained in a completely unsupervised fashion even when the number of labels is very small and the labels are noisy.  In addition, we showed that one can obtain even better results if one incorporates the labels into the learning process using a simple supervised regularizer. In particular, both unsupervised model selection and semi-supervised training are surprisingly robust to imprecise labels (inherent with human annotation) and partial labeling of factors of variation (in case not all factors can be labeled), meaning that these approaches are readily applicable in real-world machine learning systems. The findings of this paper provide practical guidelines for practitioners to develop such systems and, as we hope, will help advancing disentanglement research towards more practical data sets and tasks. “

---

### Official Review · AnonReviewer2 · 2019-10-23
**Official Blind Review #2**

**Rating:** 6

**Review:**

This paper considers the challenge of learning disentangled representations---i.e. learning representations of data points x, r(x), that capture the factors of variation in an underlying latent variable z that controls the generating process of x---and studies two approaches for integrated a small number of data points manually labeled with z (or noisier variants thereof): one using these to perform model selection, and another incorporating them into unsupervised representation learning via an additional supervised loss term.  This investigation is motivated by recent results concluding that inductive biases are needed otherwise learning disentangled representations in an unsupervised fashion is impossible.  The paper poses its overall goal as making this injection of inductive biases explicit via a small number (~100 even) of (potentially noisy) labels, and reports on exhaustive experiments on four datasets.

I think that this paper merits acceptance because (a) the motivation of taking a necessity in practice (somehow selecting models / injecting inductive biases) and making it more explicit in the approach is a good one, and because the thorough empirical survey (and simple, but novel, contribution of a new semi-supervised representation learning objective) are likely valuable contributions to this community.

One negative comment overall would be that the results are not that surprising: that is, the fact that using labels either (a) to do model validation or (b) in a semi-supervised fashion would help is not too surprising.  However, I believe in the context of (a) making more explicit a practical (and theoretically) necessary step in the pipeline of learning representations, and (b) contributing a comprehensive empirical study, this is a worthwhile contribution.

Minor notes:
- Fig. 1 isn't the most intuitive- ideally would be better explained for the headlining figure
- 8.57 P100 GPU years is ~= $75k based on a cursory glance at cloud instance pricing at monthly rates... this is a lot to reproduce these experiments...

**Experience Assessment:**

I have read many papers in this area.

**Review Assessment: Checking Correctness Of Derivations And Theory:**

I assessed the sensibility of the derivations and theory.

**Review Assessment: Checking Correctness Of Experiments:**

I assessed the sensibility of the experiments.

**Review Assessment: Thoroughness In Paper Reading:**

I read the paper at least twice and used my best judgement in assessing the paper.

---

> ### Author Response · Authors · 2019-11-08
> **Answer to Reviewer 2**
>
> We thank the reviewer for their thorough feedback and insightful comments.
>
> We concur that the finding that explicit supervision is useful for disentanglement may not be too surprising but we would also argue that there are important questions on how supervision should be incorporated into the learning process:
>
> * Are disentanglement scores sample efficient and robust to imprecision?
> * Are all scores equivalent?
> * Is it better to keep all labels for validation or should they be used for training?
>
> We are not aware of any work rigorously addressing these questions. Without extensive experiments on many different data sets, it is unclear what happens especially when very few and imprecise labels are used. Training protocol and all code will be released.

---

### Official Review · AnonReviewer3 · 2019-10-24
**Official Blind Review #3**

**Rating:** 6

**Review:**

The paper presents evidence that even a tiny bit of supervision over the factors of variation in a dataset presented in the form of semi-supervised training labels or for unsupervised model selection, can result in models that learn disentangled representations. The authors perform a thorough sweep over multiple datasets, different models classes and ways to provide labeled information. Overall, this work is a well executed and rigorous empirical study on the state of disentangled representation learning. I think experimental protocol and models trained models will prove extremely useful for future work and advocate for accepting this paper.

Comments

1) Would it be possible to use the few labeled factors of variation in a meta-learning setup rather than as a regularizer?

2) The paper provides high level conclusions about the impact of having supervised model selection or semi-supervised learning in models in general, but doesn’t offer much discussion into their behavior under specific settings (i.e.) it seems to be hard to pick a winner amongst presented model. Some are better with 100 labeled examples but don’t scale as well as others when an order of magnitude more labeled data is available. It is certainly hard to discuss all the thousands of experimental observations, but the paper can benefit from some more fine-grained analysis.

Minor
Figure 4 is hard to comprehend without a model to index mapping similar to Figure 3

**Experience Assessment:**

I have read many papers in this area.

**Review Assessment: Checking Correctness Of Derivations And Theory:**

N/A

**Review Assessment: Checking Correctness Of Experiments:**

I assessed the sensibility of the experiments.

**Review Assessment: Thoroughness In Paper Reading:**

I read the paper at least twice and used my best judgement in assessing the paper.

---

> ### Author Response · Authors · 2019-11-08
> **Answer to Reviewer 3**
>
> We thank the reviewer for their thorough feedback and insightful comments.
>
> QUESTION 1) We expect that the meta-learning approach would work:
>
> In Figure 2, we observed that it is possible to estimate the disentanglement scores reasonably well with as little as 100 noisy labeled examples (except for SAP). Further, we observed that incorporating supervision while training is beneficial regardless of the noise type (Figure 4).
>
> QUESTION 2) According to the study of Locatello et al., ICML 2019, hyperparameters and seeds are more important than the objective choice. Therefore, it is to be expected that even with limited supervision, the underlying unsupervised objective is secondary. Overall, semi-supervised training consistently outperforms supervised validation, even if we do not directly optimize for disentanglement.

---

### Author Response · Authors · 2019-11-12
**Updates:**

We made small updates to the draft, overall strengthening captions and conclusions with more direct and concise statements.

---

### Decision · Program_Chairs · 2019-12-19

**Decision:**

Accept (Poster)

**Comment:**

This paper addresses the problem of learning disentangled representations and shows that the introduction of a few labels corresponding to the desired factors of variation can be used to increase the separation of the learned representation.

There were mixed scores for this work. Two reviewers recommended weak acceptance while one reviewer recommended rejection. All reviewers and authors agreed that the main conclusion that the labeled factors of variation can be used to improved disentanglement is perhaps expected. However, reviewers 2 and 3 argue that this work presents extensive experimental evidence to support this claim which will be of value to the community. The main concerns of R1 center around a lack of clear analysis and synthesis of the large number of experiments. Though there is a page limit we encourage the authors to revise their manuscript with a specific focus on clarity and take-away messages from their results.

After careful consideration of all reviewer comments and author rebuttals the AC recommends acceptance of this work. The potential contribution of the extensive experimental evidence warrants presentation at ICLR. However, again, we encourage the authors to consider ways to mitigate the concerns of R1 in their final manuscript.